# Dual-Mode Visual System for Brain–Computer Interfaces: Integrating SSVEP and P300 Responses

**DOI:** 10.3390/s25061802

**Published:** 2025-03-14

**Authors:** Ekgari Kasawala, Surej Mouli

**Affiliations:** Engineering for Health Research Group, Biomedical Engineering, Aston University, Aston Street, Birmingham B4 7ET, UK

**Keywords:** BCI, EEG, visual stimuli, SSVEP, P300, hybrid-BCI, COB-LED, assistive technology

## Abstract

In brain–computer interface (BCI) systems, steady-state visual-evoked potentials (SSVEP) and P300 responses have achieved widespread implementation owing to their superior information transfer rates (ITR) and minimal training requirements. These neurophysiological signals have exhibited robust efficacy and versatility in external device control, demonstrating enhanced precision and scalability. However, conventional implementations predominantly utilise liquid crystal display (LCD)-based visual stimulation paradigms, which present limitations in practical deployment scenarios. This investigation presents the development and evaluation of a novel light-emitting diode (LED)-based dual stimulation apparatus designed to enhance SSVEP classification accuracy through the integration of both SSVEP and P300 paradigms. The system employs four distinct frequencies—7 Hz, 8 Hz, 9 Hz, and 10 Hz—corresponding to forward, backward, right, and left directional controls, respectively. Oscilloscopic verification confirmed the precision of these stimulation frequencies. Real-time feature extraction was accomplished through the concurrent analysis of maximum Fast Fourier Transform (FFT) amplitude and P300 peak detection to ascertain user intent. Directional control was determined by the frequency exhibiting maximal amplitude characteristics. The visual stimulation hardware demonstrated minimal frequency deviation, with error differentials ranging from 0.15% to 0.20% across all frequencies. The implemented signal processing algorithm successfully discriminated between all four stimulus frequencies whilst correlating them with their respective P300 event markers. Classification accuracy was evaluated based on correct task intention recognition. The proposed hybrid system achieved a mean classification accuracy of 86.25%, coupled with an average ITR of 42.08 bits per minute (bpm). These performance metrics notably exceed the conventional 70% accuracy threshold typically employed in BCI system evaluation protocols.

## 1. Introduction

Brain–computer interface (BCI) systems, also termed neural interfaces, establish a direct communication pathway between the Central Nervous System (CNS) and external devices through the detection, analysis, and translation of neural signals, particularly in the field of neurorehabilitation [1,2,3]. All brain–computer interface (BCI) systems adhere to a similar functional model, albeit with minor variations in their precise implementation and technical specifications. The assessment of cerebral activity can be accomplished through diverse neuroimaging and neurophysiological modalities, including functional near-infrared spectroscopy (fNIRS), electrocorticography (ECoG), functional magnetic resonance imaging (fMRI), and electroencephalography (EEG). EEG has emerged as the predominant methodology in BCI applications due to its favourable characteristics comprising economically viable instrumentation, the non-invasive nature of signal acquisition, and superior portability [4,5]. These attributes have contributed significantly to its widespread adoption in the field. These systems, whether non-invasive (utilising surface electrodes) or invasive (employing implanted electrodes), fundamentally operate through a structured sequence of signal acquisition and processing stages [6]. Initially, raw electrophysiological signals from the user’s brain activity are detected through electrode arrays (typically ranging from microvolts to millivolts in amplitude) and digitised via analogue-to-digital conversion before transmission to a computing system. These signals then undergo pre-processing protocols to minimise physiological and environmental artefacts (such as electromagnetic interference, muscle activity, and ocular movements) and enhance the signal-to-noise ratio (SNR) through spatial and temporal filtering techniques [7].

Specific signal characteristics are isolated during the feature extraction phase, including amplitude variations, frequency band powers, and temporal patterns. These features undergo classification through computational algorithms to identify distinct brain activity patterns associated with intended user commands. The classified signals are then decoded and translated into control inputs for the external interface, generating real-time feedback to the user through visual, auditory, or tactile channels, at which point the acquisition-processing cycle recommences [8,9]. The systematic workflow of these processes is depicted in Figure 1.

Multiple EEG paradigms have emerged as an improvement over traditional BCI platforms, which predominantly relied on singular EEG paradigms [10]. The limitation of single-paradigm systems lies in their potential incompatibility with certain users and their susceptibility to erroneous signal recognition. Recent advancements in BCI systems have demonstrated enhanced performance metrics through the integration of multiple paradigms, yielding superior accuracy and increased response rates in the control of peripheral applications [11,12].

BCI systems principally utilise EEG signals derived from steady-state visual-evoked potentials (SSVEP) or transient-evoked potentials, particularly the P300 component [13,14]. The preferential employment of these neurophysiological paradigms stems from their demonstrated capacity to achieve superior recognition accuracy in practical applications. SSVEP-based paradigms have garnered substantial adoption in BCI systems owing to their advantageous characteristics, including elevated information transfer rates (ITR), robust signal-to-noise ratios (SNR), rapid response latency, and minimal user training requirements. SSVEPs manifest as periodic neural responses elicited by rhythmic photic stimulation at predetermined frequencies [15,16]. These responses exhibit frequency-locked oscillations corresponding to both the fundamental frequency of the visual stimulus and its harmonic components. This phenomenon, termed frequency tagging, involves the presentation of distinct visual stimuli oscillating at known frequencies. Upon attentional engagement by the subject, these stimuli generate frequency-specific SSVEP responses that can be quantitatively analysed following neural signal acquisition and digitalisation [17].

Although SSVEP-based BCIs exhibit advantageous characteristics, they present several significant limitations. The photic stimulation inherent to these systems poses a risk of triggering photosensitive epilepsy, thereby restricting their applicability across the broader population due to potential adverse health implications. Prolonged exposure to SSVEP stimuli necessitates sustained visual fixation on flickering light sources, which can induce optical fatigue [18,19]. This fatigue subsequently compromises user performance and system accuracy. This fatigue can adversely affect user performance and system precision. Moreover, the SSVEP response varies amongst users, with performance fluctuating based on low or high visual stimulus flickering frequencies [19]. Individual variations in SSVEP response strength create obstacles for standardising the system for widespread adoption [20]. Furthermore, SSVEP signal strength and frequency typically decrease with age or due to neurological conditions, impacting performance [21]. Whilst researchers have investigated various methodological approaches to address these constraints, considerable scope remains for optimising both stimulus presentation paradigms and signal processing algorithms to enhance system precision and user experience for recent studies [22,23].

Studies demonstrate that LED-based visual stimuli consistently produce more robust SSVEP neural responses compared to LCD displays, due to their superior temporal precision and luminance control [21]. LEDs offer precise frequency control without refresh rate limitations, enabling the exploration of optimal stimulation frequencies (typically 6–30 Hz range), while larger stimuli enhance SSVEP amplitude by recruiting greater populations of neurons across the primary visual cortex, improving signal-to-noise ratios. Standard LCD refresh rates (60 Hz) restrict available stimulation frequencies to divisors of 60, while LED systems can generate any frequency within physiological constraints, and this understanding has led to development of hybrid systems.

Inter-stimulus distance significantly influences SSVEP performance, with greater distances improving classification accuracy [24]. Whilst lower and medium frequencies yield higher signal-to-noise ratios, they can induce visual fatigue during extended use, compromising system accuracy [19]. Higher frequencies offer enhanced comfort but potentially reduced SNR. Regarding colour, green light minimises eye strain and maintains high ITR during prolonged use, whereas red light poses risks for photosensitive epileptic seizures [25].

Conversely, P300 component manifests as a positive deflection in event-related potentials (ERPs) of human cerebral activity, occurring approximately 300 ms subsequent to the presentation of a stochastic stimulus [26,27,28]. This endogenous potential, predominantly observed in the parietal cortex, represents the cognitive processing of contextually significant stimuli within a sequence of standard events. The neurophysiological response enables BCI applications to exploit the temporal characteristics of the P300 event to deduce user intent based on the precise latency of the positive deflection. This temporal specificity facilitates the development of robust classification algorithms for real-time intent detection [29]. The P300 paradigm demonstrates information transfer rates comparable to SSVEP systems whilst requiring abbreviated training intervals, making it particularly suitable for applications demanding rapid user adaptation and consistent performance metrics [30].

Hybrid BCIs combine multiple input methods or paradigms to boost performance, integrating EEG with other physiological signals, utilising multi-sensory approaches, or merging different EEG patterns. These systems overcome individual paradigm limitations, enhancing accuracy, reliability, ITR, and user performance whilst reducing false positives. However, they incur higher costs and greater operational complexity [31].

Scientific investigations have demonstrated that BCI systems employing singular EEG paradigms exhibit diminished accuracy compared to hybrid implementations, owing to potential user incompatibility and susceptibility to erroneous signal classification [32,33]. This limitation arises from the inherent variability in individual neurophysiological responses and the complex nature of brain signal patterns. Hybrid architectures, which integrate multiple neurophysiological paradigms such as SSVEP and P300 responses, demonstrate enhanced capability in discriminating distinct cognitive intentions, whilst simultaneously reducing response latency and elevating information transfer rates. Notably, Bai et al. [34] achieved 94.29% accuracy and 28.64 bits/min ITR in their speller system, whilst Kapgate et al. [12,35] demonstrated strong accuracy and viability in virtual reality gaming and avatar control applications. The synergistic integration of complementary paradigms facilitates robust signal processing and classification methodologies. These improved performance metrics, encompassing accuracy, speed, and reliability, underscore the advantages of multi-paradigmatic approaches in contemporary BCI system design, particularly for applications requiring precise user intent detection and reliable command execution.

This research developed and tested a hybrid SSVEP + P300 BCI system for improved classification accuracy and reliability. The system utilised a portable dual-stimulus design, enabling sequential validation of user intention. Primary classification employed Power Spectral Density analysis for SSVEP frequency identification, whilst P300 event markers provided secondary verification to minimise false positives. The design prioritised computational efficiency and practical usability, making it suitable for real-world applications requiring robust command verification.

## 2. Materials and Methods

### 2.1. Hardware Design

The visual stimuli comprised a geometrically optimised array of eight light-emitting diodes (LEDs), specifically engineered to maximise visual-evoked potential amplitude and signal quality. The primary stimulation elements consisted of four radially arranged green COB (Chip on Board) LEDs, each with a diameter of 80 mm (wavelength: 520–530 nm), selected for SSVEP elicitation due to the heightened photoreceptor sensitivity and superior cortical response [25,36,37]. Four high-power 1-watt red LEDs (wavelength: 620–625 nm) were concentrically positioned within the COB array to facilitate P300 event-related potential responses.

The precision control architecture was implemented via a Teensy 3.2 microcontroller, PJRC, Sherwood, Portland, OR, USA, Sourced from Mouser UK (Buckinghamshire, UK) featuring an ARM Cortex-M4 processor operating at 72 MHz clock frequency. The system employed a multithreaded architecture to generate precisely timed parallel outputs, enabling the simultaneous control of four LEDs at distinct frequencies (7 Hz, 8 Hz, 9 Hz, and 10 Hz) for SSVEP elicitation, as shown in Figure 2. Each frequency was generated with a base timing resolution of 13.89 nanoseconds, ensuring precise phase control and temporal stability crucial for reliable steady-state visual-evoked responses. The multithreaded architecture ensured deterministic timing through independent thread execution, maintaining precise frequency generation and phase relationships between stimuli, which are essential for optimal SSVEP response discrimination.

For P300 event-related potential elicitation, four red LEDs were programmed with a pseudorandom stimulus presentation protocol, with each LED generating a single flash at random intervals within a 2000 ms epoch. Each LED flash event was temporally marked through ASCII character transmission (‘o’, ‘p’, ‘q’, ‘r’) via serial communication (baud rate: 9600 bits/s), enabling precise temporal synchronisation between stimulus onset and electroencephalographic data acquisition. This configuration enabled the precise temporal marking of stimulus events for subsequent P300 component extraction and analysis, with each marker uniquely identifying the corresponding LED stimulus source. Figure 3 illustrates the SSVEP and P300 hybrid stimuli.

Serial data transmission between the Teensy microcontroller and the host computer was implemented via an FTDI FT232R USB-to-UART interface controller, sourced from Mouser UK, achieving reliable event marker transmission with a latency of less than 1 ms. This FTDI-based serial interface enabled precise temporal synchronisation between stimulus presentation events and EEG data acquisition through microsecond-level temporal resolution, essential for accurate event-related potential analysis.

### 2.2. Signal Acquisition and Processing

EEG signal acquisition was performed using a g.tec Unicorn Hybrid Black (www.gtec.at) wireless amplification system (sampling rate: 250 Hz, resolution: 24-bit). Six electrodes were positioned according to the International 10–20 system, comprising three midline locations: frontal (Fz), central (Cz), and parietal (Pz); two parieto-occipital sites: left (PO7) and right (PO8); and midline occipital (Oz). Electrode-scalp impedances were maintained below 5 kΩ through the application of the conductive gel. The EEG was recorded with reference to the left mastoid electrode and a ground electrode positioned at AFz. Signal acquisition utilised wireless data transmission. A schematic representation of the complete hardware, including the visual stimuli, data acquisition hardware, and robotic control system, is presented in Figure 4. The hybrid stimuli implementation workflow is shown in Figure 5.

Signal processing protocols incorporated sequential filtering operations: initially, a 50 Hz notch filter was applied to eliminate power line interference from the raw EEG data. Subsequently, signal-specific filtering was implemented, SSVEP data underwent bandpass filtering (6.5–30 Hz, Butterworth, 4th order), and P300 data were processed using a 15 Hz low-pass filter (Butterworth, 4th order). For SSVEP feature extraction, power spectral density (PSD) estimation was performed using Welch’s method (Hamming window, 50% overlap), with maximum amplitude values identified at the target frequencies (7, 8, 9, and 10 Hz). P300 response analysis involved the temporal alignment of event markers with their corresponding timestamps, followed by peak detection within a 290–500 ms post-stimulus window. This window was selected to encompass the characteristic P300 component latency range for visual stimuli.

### 2.3. Methodological Validation of Experimental Design

Study participants were recruited according to predefined inclusion and exclusion criteria. The inclusion criteria specified individuals aged 18 years or above with no prior BCI experience. Exclusion criteria comprised any history or diagnosis of photosensitive epilepsy, ensuring participant safety during visual stimulation protocols. The final cohort comprised 12 participants (7 female, 5 male; mean age = 21.0 years) all with normal or corrected-to-normal vision.

The visual stimulation apparatus was positioned at a fixed distance of 60 cm from the participant’s nasion, corresponding to a visual angle of approximately 5° for each LED. Participants were instructed to maintain a fixed gaze on individual LEDs sequentially, with transitions between stimuli guided by auditory cues. Five experimental trials were conducted per participant, with mandatory five minute rest intervals between trials to minimise visual fatigue and maintain optimal attention levels. All trials were conducted under controlled ambient illumination (approximately 250 lux) to ensure consistent visual stimulus contrast.

All experimental procedures adhered to the ethical principles established by the World Medical Association’s Declaration of Helsinki (2013) for human participant research. The study protocol, participant information sheets, and consent documentation received formal approval from the Research Ethics Committee at Aston University. All participants were provided with comprehensive written and verbal information regarding the experimental procedures, and written informed consent was obtained before study participation.

Output control implementation utilised a LEGO^®^ MINDSTORMS^®^ EV3 robotic platform, with directional navigation (forward, backward, left, and right) determined by the processed EEG signals. Control decisions were based on two criteria: maximum amplitude detection in the SSVEP frequency spectrum and P300 event-related potential peak identification within the designated temporal window (290–500 ms post-stimulus). Upon successful feature extraction and classification, command signals were transmitted to the EV3 robotic platform via Bluetooth protocol. The mapping between extracted neurophysiological features and corresponding robotic directional commands is presented in Table 1. Real-time auditory feedback was implemented through the EV3’s integrated speaker system, with successful command execution indicated by a single auditory pulse (frequency: 1 kHz, duration: 200 ms) and failed command execution signalled by a double auditory pulse (frequency: 1 kHz, duration: 200 ms, inter-pulse interval: 100 ms).

## 3. Results

To validate the frequency detection algorithms, power spectral density (PSD) analysis was performed on the EEG data during isolated visual stimulation from each LED. Spectral analysis was conducted using Welch’s periodogram method to identify the dominant frequency components in the EEG signal. Maximum amplitude values were extracted within frequency bands of interest (7 ± 0.5 Hz, 8 ± 0.5 Hz, 9 ± 0.5 Hz, and 10 ± 0.5 Hz) to verify the accurate detection of the stimulus frequencies and validate the signal processing pipeline. The resultant power spectral density distributions and frequency-specific response characteristics are presented in Figure 6 and Figure 7. Additionally, Figure 8 presents the P300 event marker alongside its corresponding timestamp on an EEG stream.

System performance validation was conducted through a quantitative assessment of directional command accuracy across all trials and participants. Command accuracy was defined as the successful correlation between the participant’s intended directional input (indicated by focused attention on the corresponding LED stimulus) and the resultant robotic platform movement. For each participant (n=12), directional accuracy metrics were computed per trial (5 trials) for all four navigational commands (forward, backward, left, and right). A successful trial was recorded when the EV3 robotic platform executed the correct directional movement corresponding to the participant’s attended LED stimulus. The classification accuracy was calculated as the ratio of successful commands to total command attempts, expressed as a percentage.

The comprehensive analysis of system performance, comprising directional command accuracies across five experimental trials for all participants (n=12), including individual and mean performance metrics with corresponding standard deviations, is summarised in Table 2. An analysis of individual participant performance metrics, as illustrated in Figure 9, reveals notable inter-subject variability in classification accuracy.

The data demonstrate that participants S9 and S11 achieved superior performance (>95% accuracy), whilst participants S1–S8 and S10–S12 exhibited accuracies ranging between 75% and 90%. This heterogeneity in performance may be attributed to several factors, including variations in individual photoreceptor sensitivity, cognitive attention levels, and neurophysiological response characteristics. Notably, all participants (n=12) maintained classification accuracies above the established 70% threshold criterion for practical BCI implementations, substantiating the robustness of the dual-mode paradigm across a diverse user cohort. These findings suggest that individual differences in visual processing and cognitive engagement significantly influence BCI performance outcomes.

## 4. Discussion

The empirical findings substantiate the effectiveness of the proposed dual-modality visual stimulation hardware, which integrates steady-state visual-evoked potentials and P300 event-related potentials. The system achieved a mean classification accuracy of 86.25% across the participant cohort (n=12), markedly surpassing the established 70% threshold criterion conventionally employed in brain–computer interface implementations. Spectral analysis via power density estimation validated the precise generation and detection of the four designated stimulation frequencies (7 Hz, 8 Hz, 9 Hz, and 10 Hz). Oscilloscopic measurements demonstrated negligible frequency deviation ranging from 0.15% to 0.20%, substantiating the robustness of the hardware architecture. The quantitative assessment revealed that forward and backward commands (7 Hz and 9 Hz) showed marginally higher accuracy than left and right directional controls, as shown in Figure 10.

The observed 100% accuracy for forward and back movements, contrasted with variations in left and right accuracy, likely stemmed from the visual stimulus positioning and viewing angles. The experimental setup placed stimuli 60 cms from users, with 8.9° vertical viewing angles for vertical LEDs (forward/back movements) and 13.4° horizontal angles (left/right movements). Whilst the design considered peripheral vision and aligned with research indicating optimal stimuli frequency interference reduction between 4–13-degree angles [38], the asymmetry in horizontal viewing angles may have introduced perceptual variations. These findings suggest scope for optimisation through refined stimulus placement and systematic investigation of how subtle angular variations affect BCI control performance across directional movements.

The temporal progression of classification accuracy, as illustrated in Figure 11, reveals a noteworthy pattern wherein performance metrics demonstrate consistent improvement from session 1 (79.17%) through session 4 (91.67%), followed by a marginal decline in session 5 (89.58%). This observed degradation in performance during the final session, despite implemented rest intervals, suggests the manifestation of visual fatigue, a known phenomenon in SSVEP-based paradigms. This deterioration is likely attributable to prolonged exposure to repetitive visual stimulation, manifesting in decreased photoreceptor sensitivity and diminished cortical responses. Notably, despite these constraints, all participants (n=12) maintained classification accuracies above the established 70% threshold criterion for practical BCI implementations, substantiating the robustness of the dual-mode paradigm across a diverse user cohort. These findings underscore the critical importance of optimising session durations and rest protocols in practical BCI implementations, particularly for applications requiring extended periods of user engagement. Future investigations would benefit from incorporating physiological markers of visual fatigue and implementing adaptive stimulus parameters to maintain consistent performance across extended operational periods.

## 5. Conclusions

This investigation has successfully demonstrated the efficacy of a novel dual-mode visual stimulation system that integrates SSVEP and P300 responses for brain–computer interface applications. The system achieved a mean classification accuracy of 86.25% across participants, notably exceeding the conventional 70% threshold for practical BCI implementations, whilst maintaining minimal frequency deviation (0.15–0.20%) in stimulus generation. The technical validation confirmed the successful integration of LED-based SSVEP stimulation (7–10 Hz) with P300 event markers, supported by robust signal processing for concurrent feature extraction. An analysis of user performance revealed consistent improvement across initial sessions, though the impact of visual fatigue became apparent in extended use scenarios. Despite these fatigue effects, the system maintained acceptable accuracy levels throughout testing. Future development should focus on implementing adaptive stimulus parameters, optimising session durations and rest protocols, incorporating physiological markers of fatigue, and enhancing system robustness for extended operational periods. These results demonstrate that this hybrid approach offers a promising direction for practical BCI applications, particularly in assistive technology and device control scenarios, providing enhanced reliability and accuracy compared to single-modality systems whilst maintaining user accessibility.

## Figures and Tables

**Figure 1 sensors-25-01802-f001:**
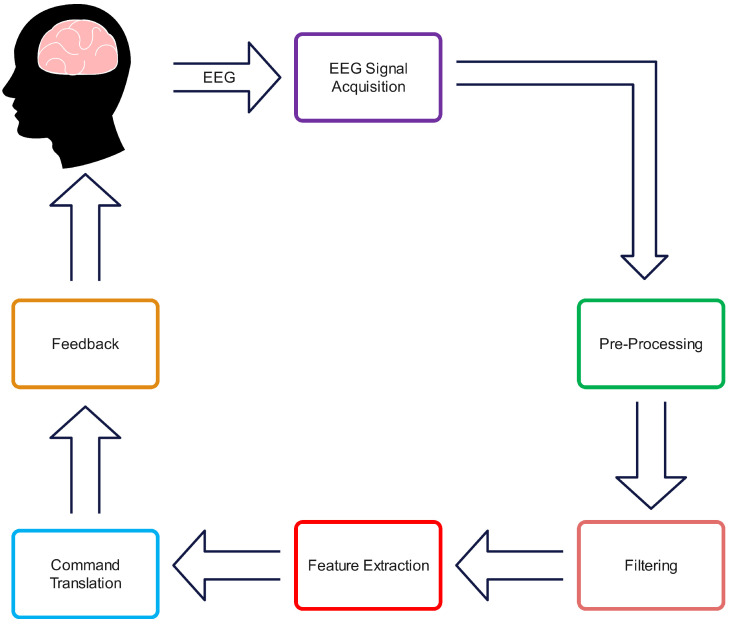
Fundamental components of BCI system.

**Figure 2 sensors-25-01802-f002:**
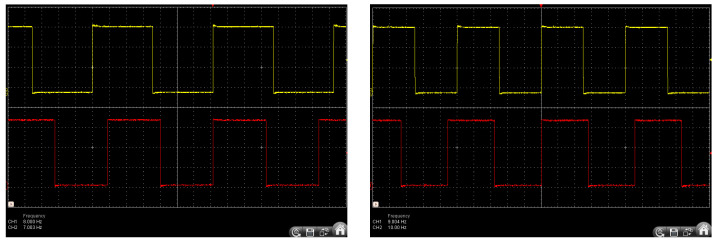
Oscilloscopic validation of LED Stimuli frequencies.

**Figure 3 sensors-25-01802-f003:**
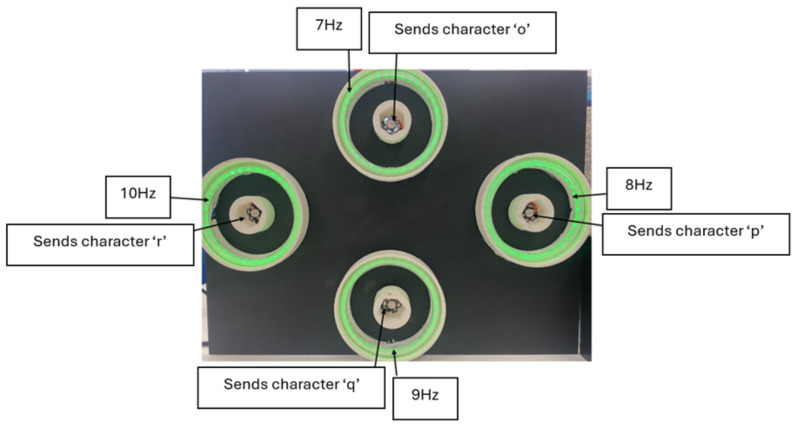
SSVEP and P300 event markers for hybrid stimuli.

**Figure 4 sensors-25-01802-f004:**
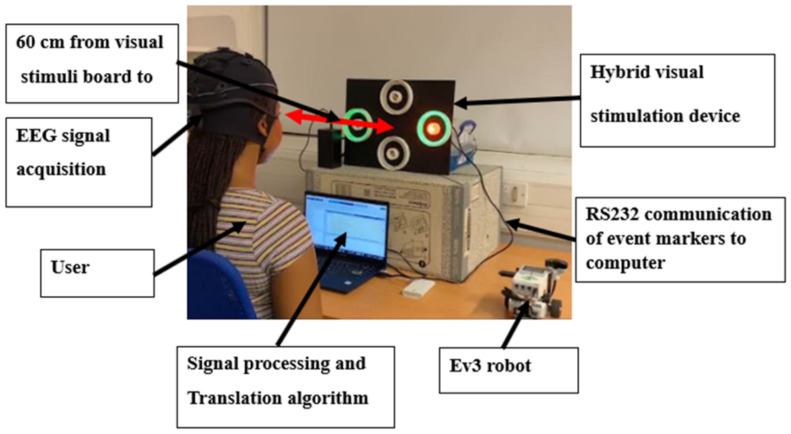
Prototype BCI real-time system for robot direction control.

**Figure 5 sensors-25-01802-f005:**
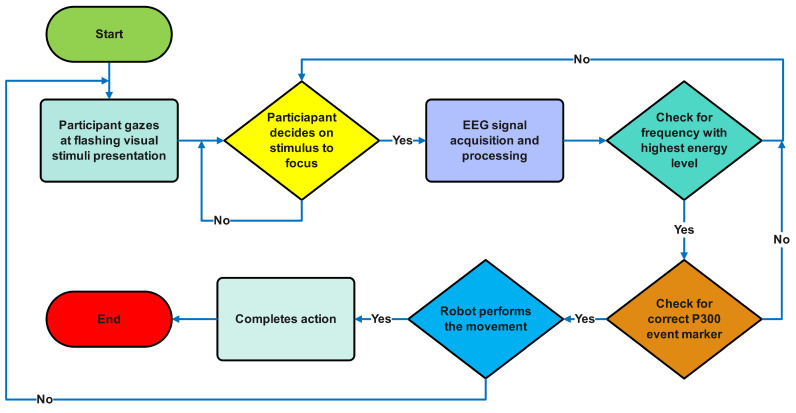
Hybrid stimuli implementation workflow.

**Figure 6 sensors-25-01802-f006:**
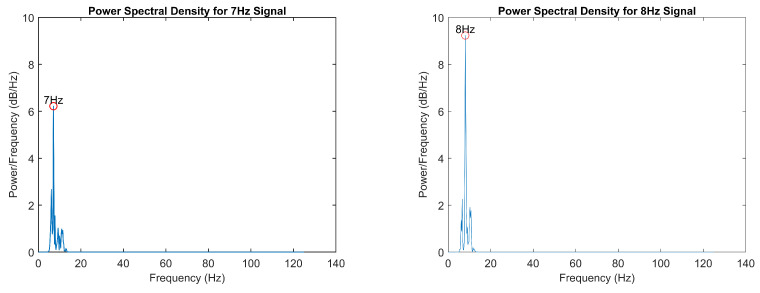
Power spectral density analysis of 7 Hz and 8 Hz.

**Figure 7 sensors-25-01802-f007:**
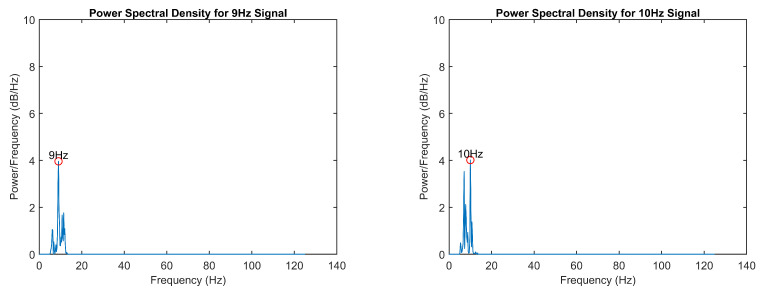
Power spectral density analysis of 9 Hz and 10 Hz.

**Figure 8 sensors-25-01802-f008:**
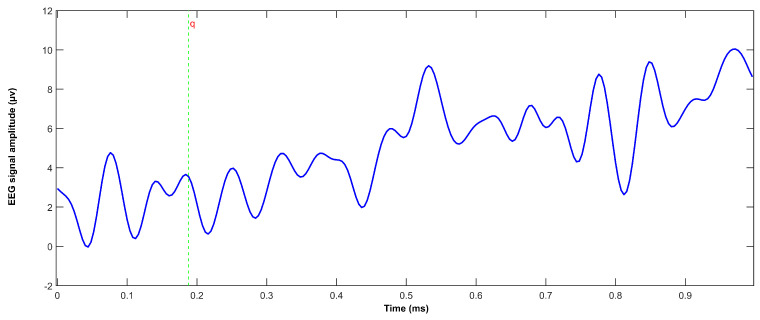
P300 potential after event marker ‘q’ for backward direction.

**Figure 9 sensors-25-01802-f009:**
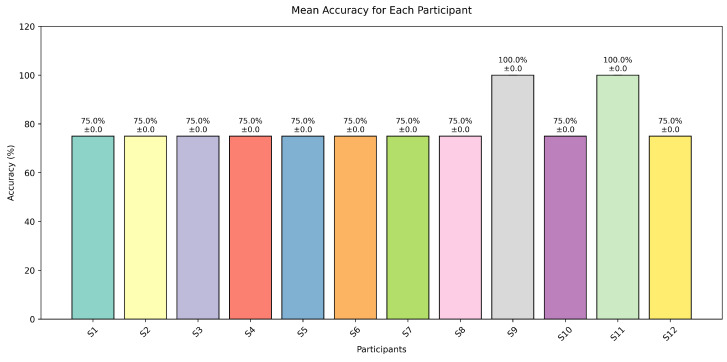
BCI control accuracy across participants.

**Figure 10 sensors-25-01802-f010:**
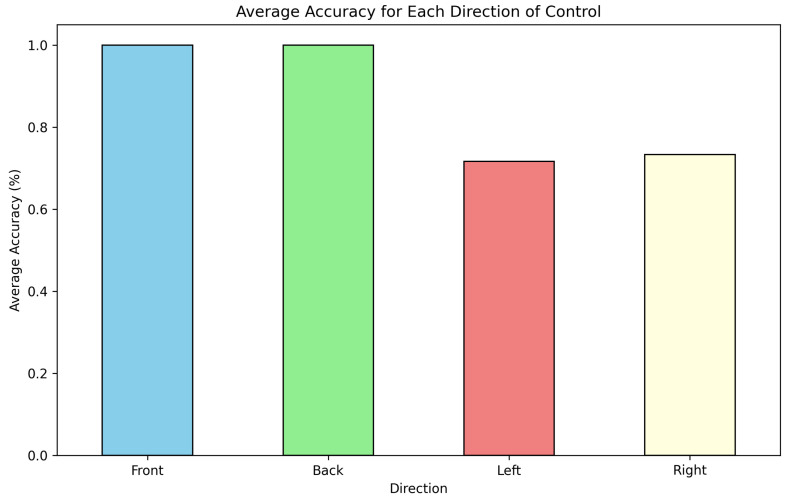
Average accuracy for each direction of control.

**Figure 11 sensors-25-01802-f011:**
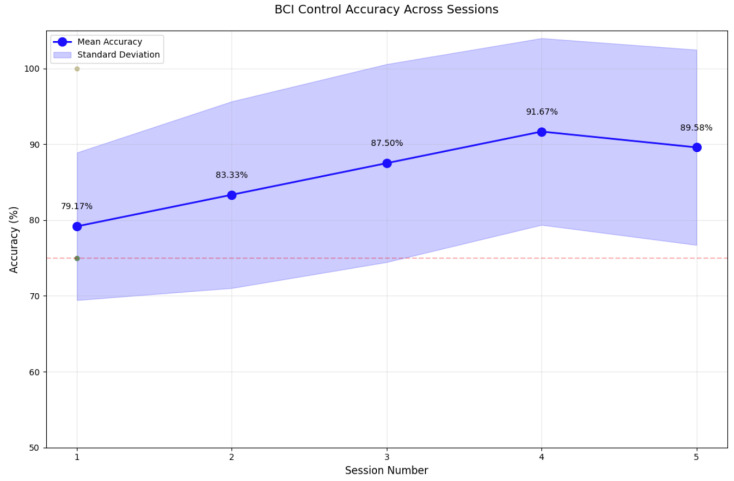
BCI control accuracy across sessions.

**Table 1 sensors-25-01802-t001:** Robot direction and corresponding features.

SSVEP Frequency (Hz)	P300 Event Marker	Robot Navigation
7	o	Forward
8	p	Right
9	q	Backward
10	r	Left

**Table 2 sensors-25-01802-t002:** BCI control performance data for all participants.

Participant 1						Participant 2					
Trial	F	B	L	R	A (%)	Trial	F	B	L	R	A (%)
1	1	1	1	0	75	1	1	1	1	0	75
2	1	1	0	1	100	2	1	1	0	1	75
3	1	1	1	1	100	3	1	1	1	1	100
4	1	1	1	1	75	4	1	1	1	1	100
5	1	1	1	1	100	5	1	1	1	0	75
Participant 3						Participant 4					
1	1	1	1	0	75	1	1	1	1	0	75
2	1	1	0	1	75	2	1	1	1	0	75
3	1	1	1	1	100	3	1	1	1	1	100
4	1	1	1	1	100	4	1	1	1	1	100
5	1	1	0	1	75	5	1	1	0	1	75
Participant 5						Participant 6					
1	1	1	0	1	100	1	1	1	1	0	75
2	1	1	1	0	75	2	1	1	1	1	100
3	1	1	1	1	75	3	1	1	0	1	75
4	1	1	1	1	100	4	1	1	0	1	75
5	1	1	1	1	100	5	1	1	1	1	100
Participant 7						Participant 8					
1	1	1	1	0	75	1	1	1	1	0	75
2	1	1	1	1	100	2	1	1	0	1	75
3	1	1	1	1	100	3	1	1	1	1	100
4	1	1	1	1	100	4	1	1	1	1	100
5	1	1	0	1	75	5	1	1	1	1	100
Participant 9						Participant 10					
1	1	1	1	1	100	1	1	1	0	1	75
2	1	1	1	1	100	2	1	1	1	1	100
3	1	1	0	1	75	3	1	1	1	1	100
4	1	1	1	0	75	4	1	1	0	1	75
5	1	1	1	1	100	5	1	1	1	0	75
Participant 11						Participant 12					
1	1	1	1	1	100	1	1	1	1	0	75
2	1	1	0	1	75	2	1	1	1	0	75
3	1	1	0	1	75	3	1	1	1	1	100
4	1	1	1	1	100	4	1	1	1	0	100
5	1	1	1	1	100	5	1	1	1	1	100

F = Front, B = Back, L = Left, R = Right, and A = Accuracy. 1 indicates successful control, and 0 indicates unsuccessful control in the respective direction.

## Data Availability

Data are contained within the article.

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
