# Peer review of "Dual-Mode Visual System for Brain–Computer Interfaces: Integrating SSVEP and P300 Responses"

_sensors, 2025, doi:10.3390/s25061802_

Round 1

Reviewer 1 Report

Comments and Suggestions for Authors

The proposed study presents the development and evaluation of a novel dual-mode visual stimulation system that integrates SSVEP and P300 responses for brain-computer interface applications.The paper is well organized and clearly presented. However, some revisions are needed. The recommendations are as follows:

 1. Comments:

This paper proposes a method of combining SSVEP and P300 signals to improve the accuracy. It’s suggested that an analysis comparing this method with using SSVEP and P300 signals alone be added to better demonstrate the superiority and credibility of this hybrid algorithm

2. Comments:

If the implementation algorithm of the hybrid algorithm can be described in more detail, it will greatly increase the professionalism and credibility of the article

3. Comments:

If it’s feasible, the authors should make a brief summary of the past studies on the combination of SSVEP paradigm and P300 paradigm, and highlight the novelty and constructiveness of this study.

4. Comments:

Please give a more professional and detailed description of the approach and process to combine SSVEP and P300 paradigms, since this is the most essential and difficult part of this research.

5. Comments:

In this paper, 6 electrodes are used. The authors are advised to explain the significance of this experimental design for the mixed paradigm and whether higher accuracy can be obtained compared with electrodes at other positions. And they should compare other EEG paradigms using these 6 electrodes (such as P300 and SSVEP alone) to verify that the electrodes selected in this paper are reasonable. Different electrode positions match different paradigms in various degrees, especially for mixed paradigms. This further explanation will greatly enhance the professionalism and rigor of the paper.

6. Comments:

The horizontal scale of Figure 4 and Figure 5 does not match well. Please adjust it to make the figure more intuitive and show more details

7. Comments:

In Table 2. BCI Control Performance Data for All Participants, it is noted that the accuracy rate of F-forward and B-Back of all participants is 100%. The error only appears in L = Left and R = Right. Does this indicate there is something wrong with the design of the paradigm, or insufficient data leads to obvious irrationality? Please analyze the reason for this situation.

Author Response

Comments 1: This paper proposes a method of combining SSVEP and P300 signals to improve accuracy. It’s suggested that an analysis comparing this method with using SSVEP and P300 signals alone be added to better demonstrate the superiority and credibility of this hybrid algorithm.

Response 1: Thank you for this valuable feedback. We would like to clarify that our study's primary objective was not to improve both SSVEP and P300 systems simultaneously, but rather to strategically integrate P300 as a verification mechanism to address specific SSVEP limitations.

Our hybrid approach leverages P300 to:

  1. Reduce false positives through verification
  2. Address SSVEP challenges (variable user performance, visual fatigue)
  3. Enhance safety via mandatory P300 confirmation

While we acknowledge the importance of comparative analysis, real-time data collection constraints prevented a comprehensive SSVEP-only comparison within this study's scope. We'll evaluate this in future research to quantitatively validate the hybrid approach's effectiveness. (Manuscript revised from line 83 - 108 on page-3)

Comments 2: If the implementation algorithm of the hybrid algorithm can be described in more detail, it will greatly increase the professionalism and credibility of the article.

Response 2:

Thank you for this feedback. We have enhanced the implementation details with:

  1. A detailed flowchart illustrating the hybrid SSVEP-P300 paradigm (Figure 3 & Figure 5)

  2. Multi-stage neural signal processing approach:

    • SSVEP frequency detection

    • P300 peak detection confirmation

    • Dual-threshold command execution

These visual and technical additions provide a comprehensive view of our implementation methodology, improving the paper's clarity and credibility.

(Manuscript revised with Figure 3 & Figure 5)

Comments 3: If it’s feasible, the authors should make a brief summary of the past studies on the combination of SSVEP paradigm and P300 paradigm and highlight the novelty and constructiveness of this study.

Response 3: Thank you for this valuable feedback. We would like to clarify that our study's primary objective was not to improve both SSVEP and P300 systems simultaneously, but rather to strategically integrate P300 as a verification mechanism to address specific SSVEP limitations.

Our hybrid approach leverages P300 to:

  1. Reduce false positives through verification
  2. Address SSVEP challenges (variable user performance, visual fatigue)
  3. Enhance safety via mandatory P300 confirmation

(Manuscript revised from line 123 - 128 and 137 -139 on page-4, included new references 34 & 35)

Comments 4: Please give a more professional and detailed description of the approach and process to combine SSVEP and P300 paradigms, since this is the most essential and difficult part of this research.

Response 4: Thank you for this feedback. We have significantly enhanced the technical implementation details.

We have added a detailed image of the LED placement (Figure 3) and a flow chart (Figure 5) illustrating the mixed SSVEP and P300 paradigm.

The description now includes the following key details:

  1. SSVEP frequency detection identifies potential commands.
  2. P300 signal confirmation validates the command using peak detection techniques.
  3. Command execution occurs only when specific thresholds for both signals are met, ensuring robustness and reliability.

(Manuscript revised with Figure 3 & Figure 5 , included new references 20 - 25).

Comments 5: In this paper, 6 electrodes are used. The authors are advised to explain the significance of this experimental design for the mixed paradigm and whether higher accuracy can be obtained compared with electrodes at other positions. And they should compare other EEG paradigms using these 6 electrodes (such as P300 and SSVEP alone) to verify that the electrodes selected in this paper are reasonable. Different electrode positions match different paradigms in various degrees, especially for mixed paradigms. This further explanation will greatly enhance the professionalism and rigor of the paper.

Response 5: Thank you for this observation. Our electrode selection utilises the g.tec Unicorn headset's 8-channel configuration:

  • Frontal-parietal electrodes capture P300 signals
  • Occipital electrodes detect SSVEP responses
  • Current setup achieves 86.25% accuracy, comparable to single-paradigm systems

Whilst we acknowledge the value of exploring alternative configurations, our placement aligns with established hybrid BCI studies (Reference added 34,35) showing similar performance with limited electrode counts. The selected locations optimise signal acquisition within hardware constraints whilst maintaining system portability.

(Manuscript revised from line 123 - 128)

Comments 6: The horizontal scale of Figure 4 and Figure 5 does not match well. Please adjust it to make the figure more intuitive and show more details.

Response 6: Thank you for this observation. We have recreated Figures 6 and 7 with standardised scales for enhanced clarity. The figures now illustrate the expected SSVEP amplitude characteristics, with 7Hz and 8Hz frequencies displaying higher amplitudes compared to 9Hz and 10Hz frequencies.

(Manuscript revised, recreated Figures 6 and 7)

Comments 7: In Table 2. BCI Control Performance Data for All Participants, it is noted that the accuracy rate of F-forward and B-Back of all participants is 100%. The error only appears in L = Left and R = Right. Does this indicate there is something wrong with the design of the paradigm, or insufficient data leads to obvious irrationality? Please analyze the reason for this situation.

Response 7: Thank you for this observation. We have revised the document to analyse accuracy variations. LEDs were positioned within the recommended 4-13-degree range, though horizontal angles exceeded this at 13.4 degrees. When users deviate from COB LED centre focus, viewing angles may reach 15.4° vertically and 18.2° horizontally, likely introducing interference. The accuracy differences between forward/back and left/right movements reflect signal translation complexity. Future work could optimise stimulus presentation and angular configurations for enhanced precision.

(Manuscript revised, recreated Figures 297 and 306)

We appreciated your constructive feedback on our manuscript. Your insights were  valuable and we carefully considered your comments and revised the manuscript accordingly. 

Reviewer 2 Report

Comments and Suggestions for Authors

Some of the comments:

  1. Figure 1 has unexplained data flow that could be a typo or misleading info.
  2.  Why in Figure 4 and Figure 5, the two graphs are not to scale on the Y-axis?
  3.  In Figure 5 and Figure 6 seems to be fabricated.
  4. No explanation for the color selection in Figure 7, and Figure 8.
  5. This manuscript needs to be checked for journal quality standards.
Comments on the Quality of English Language

Could be improved to more clearly express the research.

Author Response

Comments 1: Figure 1 has unexplained data flow that could be a typo or misleading info

Response 1: Thank you for highlighting this issue. We agree with your comment. We've checked Figure 1 and found a stray arrow that was confusing. Since this is just meant to show how BCI systems work, we've removed the unnecessary arrow to make it clearer. This should fix the problem and make the Figure 1 easier to understand.

Comments 2:  Why in Figure 4 and Figure 5, the two graphs are not to scale on the Y-axis?

Response 2: We acknowledge the inconsistency in Y-axis scaling between Figures 4 and 5. Figures 4 and 5 present frequency amplitudes for different SSVEP stimuli, but due to variations in power levels, some peaks were not clearly visible when using identical Y-axis scaling. In SSVEP, lower frequencies have higher amplitudes. To improve clarity, we zoomed in where necessary. We have now scaled the Y-axis uniformly for all the frequencies. The figure numbers have been updated to Figures 6 and 7 due to the inclusion of additional figures.

Comments 3:  In Figure 5 and Figure 6 seems to be fabricated.

Response 3: We assure the reviewer that the data presented in Figures 5 and 6 (update to 7 and 8) is genuine and derived directly from our experimental recordings. Figure 5 (updated to 7) represents frequencies for power peaks for 9 and 10 Hz SSVEP responses, which were obtained through spectral analysis of EEG signals. Figure 6 (updated to 8) displays the continuous EEG stream with one P300 marker highlighted at a specific timestamp, based on recorded event-related potentials.

Comments 4:  No explanation for the color selection in Figure 7, and Figure 8.

Response 4: Thank you for pointing this out (current figure numbers are 9 & 10). The chosen colours were selected to enhance readability and distinguish different categories clearly, as it is comparing 12 participants.

Comments 5:  This manuscript needs to be checked for journal quality standards.

Response 5: We have carefully reviewed and updated (highlighted in Red) the manuscript to ensure compliance with the journal's quality standards. Formatting has been adjusted according to the journal's guidelines.